# Characterization of the Evolutionary Pressure on *Anisodus tanguticus* Maxim. with Complete Chloroplast Genome Sequence

**DOI:** 10.3390/genes13112125

**Published:** 2022-11-15

**Authors:** Dangwei Zhou, Furrukh Mehmood, Pengcheng Lin, Tingfeng Cheng, Huan Wang, Shenbo Shi, Jinkui Zhang, Jing Meng, Kun Zheng, Péter Poczai

**Affiliations:** 1The College of Pharmacy, Qinghai Nationalities University, Xining 810007, China; 2Key Laboratory of Adaptation and Evolution of Plateau Biota (AEPB), Northwest Institute of Plateau Biology, Chinese Academy of Sciences, Xining 810008, China; 3Department of Biochemistry, Faculty of Biological Sciences, Quaid-i-Azam University, Islamabad 45320, Pakistan; 4Department of Biochemistry, Faculty of Sciences, University of Sialkot, Daska Road, Punjab 51040, Pakistan; 5Faculty of Biological and Environmental Sciences, University of Helsinki, FI-00014 Helsinki, Finland

**Keywords:** *Anisodus tanguticus* Maxim., chloroplast genome, comparative analysis, positive selection, phylogenetic relationship, hyoscyameae tribe

## Abstract

*Anisodus tanguticus* Maxim. (Solanaceae), a traditional endangered Tibetan herb, is endemic to the Qinghai–Tibet Plateau. Here, we report the de novo assembled chloroplast (cp) genome sequences of *A. tanguticus* (155,765 bp). The cp contains a pair of inverted repeated (IRa and IRb) regions of 25,881 bp that are separated by a large single copy (LSC) region (86,516 bp) and a small single copy SSC (17,487 bp) region. A total of 132 functional genes were annotated in the cp genome, including 87 protein-coding genes, 37 tRNA genes, and 8 rRNA genes. Moreover, 199 simple sequence repeats (SSR) and 65 repeat structures were detected. Comparative plastome analyses revealed a conserved gene order and high similarity of protein-coding sequences. The *A. tanguticus* cp genome exhibits contraction and expansion, which differs from *Przewalskia tangutica* and other related Solanaceae species. We identified 30 highly polymorphic regions, mostly belonging to intergenic spacer regions (IGS), which may be suitable for the development of robust and cost-effective markers for inferring the phylogeny of the genus *Anisodus* and family Solanaceae. Analysis of the Ka/Ks ratios of the Hyoscyameae tribe revealed significant positive selection exerted on the cemA, rpoC2, and clpP genes, which suggests that protein metabolism may be an important strategy for *A. tanguticus* and other species in Hyoscyameae in adapting to the adverse environment on the Qinghai–Tibetan Plateau. Phylogenetic analysis revealed that *A. tanguticus* clustered closer with *Hyoscyamus niger* than *P. tangutica.* Our results provide reliable genetic information for future exploration of the taxonomy and phylogenetic evolution of the Hyoscyameae tribe and related species.

## 1. Introduction

Chloroplasts (cp) in plant cells are photosynthetic organelles that play a critical role in the production of essential energy and metabolites [1]. Cps contain a 120–160 kb genome with typical conserved quadripartite regions, including an LSC region, an SSC region, two identical copies of IR regions, and approximately 110–130 encoded genes distributed in a circular genome [2,3]. As most plant species exhibit maternal inheritance in their cps, the cp genome has undergone less recombination and evolution [3,4,5]. Therefore, cp sequences are considered a vital resource of molecular genetic markers to analyze the profiling of gene distribution and their molecular phylogenetic relationships [6]. Furthermore, because of the high level of expression, the absence of post-transcriptional gene silencing and site-specific transformation [7], cps are useful in the agriculture and pharmaceutical industries. In recent decades, rapid advances in next-generation sequencing technologies along with reduced costs have enabled a better understanding of the phylogenetic status and the cp transformation technique in traditional herbs using cp genome information. *A. tanguticus* is a rare and endangered Solanaceae species found on the Qinghai–Tibetan Plateau in grasslands or meadows (2200–4200 m) [8]. According to traditional Tibetan medicine, this species includes an abundance of hyoscyamine, atropine, and total alkaloids [9,10,11] and is widely used for the treatment of asthma, epilepsy, and inflammation [12,13]. Despite the development of a domesticated plant [11], the main medicinal resource of *A. tanguticus* still relies on wild plant exploration and collection, which has led to the species becoming endangered [14]. Furthermore, as populations are self-incompatible and interspecific hybridization occurs [15], the population structure and genetic diversity of *A. tanguticus* in the Qinghai–Tibetan region should be evaluated. DNA barcoding is thought to be an effective approach for medicinal component identification and quality control of produced goods in addition to protecting consumers [16,17]. Previous phylogenetic studies of Solanaceae plants have been based mostly on a few cp genes or ITS sequences [18,19]. There is currently limited cp genome information of *A. tanguticus*. Here, we assembled and de novo sequenced the whole cp genome of *A. tanguticus* with Illumina sequencing platforms. Additionally, cp genome annotation and structure analysis were performed and several oligonucleotide repeats, SSR, and substitutions were identified as mutation hotspots for future DNA barcoding. Further insights on *A. tanguticus* plastome evolution and its phylogenetic relationships have been revealed through comparative studies with reported cp genomes of Solanaceae species. 

## 2. Materials and Methods

### 2.1. DNA Extraction

Fresh young leaves of *A. tanguticus* were collected from Haibei station (37.48° N, 101.2° E; Alt. 3200 m), in Qinghai province. According to the manufacturer’s instructions, a DNeasy Plant MiniKit (QIAGEN, Düsseldorf, Germany) was utilized to isolate DNA from the fresh young leaves. Spectrophotometry and electrophoresis on a 1% (*w*/*v*) agarose gel were used to evaluate the purity of the DNA. DNA of sufficient integrity and purity DNA was used for library construction and sequencing with an IlluminaHiseq2500 (San Diego, CA, USA). 

### 2.2. Plastome Assembly and Annotation 

Trimmomatic V0.36 was used to remove adapters and poor-quality reads from the raw reads [20]. Furthermore, clean reads were then mapped to the database, which was constructed from all cp genome sequences reported in the NCBI based on coverage depth and similarity. The mapped reads were then assembled into contigs [21] using SOAPdenovo2. The reads were filtered with careful error correction and different k-mers (55, 87, and 121) [22]. The scaffold of the cp genome was constructed with SSPACE [23] and GapCloser was used to fill the gaps [21]. PCR amplification and Sanger sequencing were used to validate the assembly by verifying the four boundaries between single copy (SC) and inverted repeat (IR) regions of the assembled sequences. Dual Organellar GenoMe Annotator (DOGMA, http://dogma.ccbb.utexas.edu/, accessed on 15 March 2021), CPGAVAS, and manual corrections were used to annotate the full cp genomes [24,25]. The software tRNAscan-SE was used for the prediction of tRNA genes. Circular cp genome map was drawn by Organellar Genome DRAW (OGDRAW) V1.2 [26]. The correct and complete cp genome was deposited in the National Center for Biotechnology Information (NCBI) under accession number MW246825 (*A. tanguticus*). Web-based REPuter (https://bibiserv.cebitec.uni-bielefeld.de/reputer/ accessed on 15 March 2021) was used for the analysis of repeat sequences, including forward, palindrome, reverse, and complement repeats. Sequence identity was set to >90% with a minimum repeat size of 30 base pairs. Micro Satellite Identification Tool (MISA) [27] was used to identify simple sequence repeats (SSRs), with minimum repeats of mono-, di-, tri-, tetra-, penta-, and hexa-nucleotides set to 10, 5, 4, 3, and 3, respectively. Using the tool CodonW1.4.2 (http://downloads.fyxm.net/CodonW-76666.html accessed on 15 March 2021), only 53 protein-coding genes with lengths more than 300 bp were chosen for analysis of synonymous codon usage. The relative synonymous codon usage (RSCU) and overall codon usage were both analyzed. 

### 2.3. Comparative Analysis of cp Genomes 

The plastome of *A. tanguticus* was compared with the cp genomes of *Prewalskia tangutica*, *Scopolia parviflora*, *H. niger*, *Datura stramonium*, and *Nicotiana sylvestris* in Solanaceae using Mauve software to identify evolutionary events such as gene loss, duplication, and rearrangement [28]. As a reference, the annotation of N. sylvestris was selected.

### 2.4. Molecular Evolution Analysis 

The protein-coding genes of *A. tanguticus* were evaluated for synonymous (Ks) and non-synonymous (Ka) substitution rates in comparison with the six closely related Solanaceae species (*P. tangutica*, *S. parviflora*, *H. niger*, *D. stramonium*, and *N. tabacum*). Between the compared species, the corresponding functional protein-coding gene was separately aligned using MAFFT [29]. The Ka/Ks value was then determined using the NG method of computation, created by Nei and Gojobori and implemented in KaKs calculator 2.0 [30], with the settings genetic code Appendix A (bacterial and plant plastid code). In the results, some genes had Ka/Ks values of “NA”, which meant they were not relevant. When Ks = 0, this indicates no substitutions in the alignment, or 100% match. Concatenated sequences of 67 common protein-coding genes among the studied species were used in a phylogenetic analysis based on the complete cp genomes. The two sets of sequences were aligned by MAFFT [29], and the alignments were then adjusted by the Gblocks program [31]. Phylogenetic trees were created by the RAxMLversion 8.0 program using the GTRGAMMA model by the maximum likelihood (ML) approach [32]. Each branch’s bootstrap analysis was calculated using 1000 replications.

## 3. Results and Discussion 

### 3.1. Chloroplast Genome Features of A. tanguticus 

The cp genome of *A. tanguticus* is 155,765 nucleotides and has a quadripartite architecture. The genome consists of a pair of IRs (25,881 bp), a large single copy (LSC) region (86,516 bp), and a small single copy (SSC) region (17,487 bp) (Figure 1). These results are similar to other species of Solanaceae (Table 1). The cp genome’s total GC content was 37.63% and the GC content of the IR regions was 42.87%, which is higher than those of the LSC region (35.65%) and SSC region (31.93%). Similar results have also been observed in other species [32,33], which may be due to the four rRNAs and tRNAs [34,35]. The *A. tanguticus* plastome has 132 unique genes, including 87 protein-coding genes and 37 tRNA and eight rRNA genes. The cp genome contained 131 genes altogether, including 21 functional gene duplications in IR areas (Table 1). The plastome of *A. tanguticus* (87) has more protein-coding genes than *P. tangutica* (85) and *H. niger* (80) (Table 1). Among these, eight protein-coding genes (atpF, ndhA, ndhB, petD, rpoC1, rpl16, rpl2, and rps16) and six tRNAs had one intron, while three genes (ycf3, rps12, and clp2) contained two introns (Table 2). Nineteen genes, including four ribosomal (rrn4.5, rrn5, rrn16, and rrn23), seven trn (trn-UGC, trnI-CAU, trnI-GAU, trnL-CAA, trnN-GUU, trnR-ACG, and trnV-GAC), ndhB, rps7, rps12, rpl2, rpl23, ycf1, ycf2, and ycf15 had two repeats in the *A. tanguticus* plastome (Table 2). The largest intron (2502 bp) was found in the trnK-UUU gene, and it contained matK. These results are consistent with the mustard family (Brassicaceae) cp genome [35]. 

### 3.2. Codon Usage Analysis 

RSCU frequency plays an important role in reflecting mutation bias during evolution [36]. RSCU values > 1 show the use of a codon more frequently than expected and vice versa [37]. The *A. tanguticus* plastome has a total of 80,910 bp of protein-coding genes, which encodes 26,970 codons. Among these, 2867 (6%) encoded leucine, which is the most abundant amino acid, whereas only 483 (1.0%) encoded tyrosine, which is the least prevalent amino acid in the *A. tanguticus* cp genome. According to the codon-anticodon recognition patterns, 29 tRNAs had codons that matched all 20 amino acids (Figure 2, Appendix A). AT content at the first, second, and third codon positions covered 54.3%, 61.9%, and 67.8%, respectively. This phenomenon was also higher in other plants [38,39]. There were 31 codons that exhibited clearly biased usage (RSCU > 1). Except for two codons (UUG for Leu and AUG for Met), all biased codon usage bases were A/U at the third position. Similarly, the codon for Trp did not exhibit biased usage (RSCU = 1) (Figure 2, Appendix A). This result is consistent with the *N. officinale* cp genome [35]. However, unlike these species, ATG showed the highest RSCU (1.9936) value of all codons in the *A. tanguticus* cp genome. The significant evolutionary character of the genome is evident in the codon usage bias [37,38]. Our results suggest that the *A. tanguticus* cp genome has experienced a different evolution process. 

### 3.3. Repeat Sequence and SSR Analysis 

Repeat sequences are a constant feature of illegitimate recombination and slipped-strand mispairing, and they can be utilized in plant phylogenetic studies [39,40]. The *A. tanguticus* cp genome contains 39 repeats, of which 18 are direct repeat sequences, 18 are palindromic sequences, and three are reverse repeat sequences. There were no complementary repeat sequences (Figure 3A). Analyses of oligonucleotide repeats revealed that all sequences were ≥30 bp, and direct and palindromic repeats covered 46% of the total dispersed repeat sequences in the cp genome. Interestingly, there was one palindromic sequence of 25,881 bp (Figure 3A). The number of forward and palindromic sequences was close among the different types of repeats, although they still differed among the six species (Figure 3B). Further analysis of repeat length among the four species of Hyoscyameae tribe revealed that the length of 40–49 bp in forward and palindromic sequences was greater than the other two species (Figure 3C,D), and the length of 50–59 bp may have played an important role in *H. niger* cp genome rearrangement (Figure 3C,D). However, the length of 30–40 bp for forward and palindromic sequences did not differ significantly among six species. Surprisingly, the number of reverse sequences with lengths of 30–39 bp was substantially greater in the cp genomes of *A. tanguticus* and *P. tangutica*, and there were no reverse sequences in the cp genomes of *H. niger* and *D. stramonium* (Figure 3E). Previous research has discovered long repetitions of >30 bp, which may contribute to cp genome rearrangement and increase population genetic diversity [41,42]. Our findings revealed that forward and palindromic sequences with lengths of 40–49 bp may play an essential role in the Hyoscyameae tribe, while reverse sequences with lengths of 30–39 bp are more abundant in *A. tanguticus* and *P. tangutica* than in the Hyoscyameae tribe. This may suggest that the reverse sequences of *A. tanguticus* and *P. tangutica* may play a more important role in cp genome rearrangement to adapt to the alpine environment on the Qinghai–Tibet Plateau. SSRs were 1–6 bp in the cp genome and were widely dispersed in intergenic, intron, or coding regions. SSRs had a high mutation rate and diverse copy number, which confers their important value in phylogenetic and population analyses [43,44]. Using MISA, 199 SSRs were identified in the *A. tanguticus* plastome (Appendix A). Among these, there were 120 mononucleotide, eight dinucleotide, 72 trinucleotide, seven tetranucleotide, and one pentanucleotide repeats. In addition, 60.3% mononucleotide repeats were A/T, which is consistent with results from *Physalis* species [45]. There were 61 trinucleotide SSRs in the *A. tangutica* cp genome, which is similar to four species of Hyoscyameae (Figure 3F) and more than that observed in four *Physalis* species [45]. Interestingly, dinucleotide SSRs were considerably lower than that of *P. tangutica*, *S. parviflora*, and *H. niger*, which contained 43, 43, and 42 dinucleotide SSRs, respectively. Although there was one pentanucleotide repeat in *A. tanguticus*, *S. parviflora*, and *H. niger*, there were three such repeats in the *P. tangutica* cp genome (Figure 3F). Therefore, these SSR differences may be helpful for phylogenetic and evolutionary studies. Furthermore, the LSC regions had 63.3% SSR markers, which was considerably greater than that of the SSC and IR regions. Similarly, nearly all SSRs in the *A. tanguticus* cp genome were comprised of A/T, which added 62.3% to the genome’s base composition.

### 3.4. IR Contraction and Expansion

The expansion and contraction of the IR boundaries were supposed to be an evolutionary trait of plastomes, and contraction and expansion at the IR region borders were believed to explain plastome differences [46,47]. To characterize the variation at the IR boundary, we selected the nine phylogenetically related Solanaceae species (*P. peruviana*, *Iochroma ellipticum*, *Capsicum frutescens*, *Solanum lycopersicum*, *D. stramonium*, *N. sylvestris*, *H. niger*, *S. parviflora*, and *P. tangutica*). Genomic structure and gene order are conserved in these species, especially in the Hyoscyamus group (Figure 4). The IRa/SSC border for the *A. tanguticus* cp genome was found in the 3’ region of the whole ycf1 gene. This is similar to *H. niger*, *P. tangutica*, and the other species except for *S. parviflora*. Similarly, except for *S. parviflora* and *N. sylvestris*, *A. tanguticus* and the other seven species contained the ycf1 pseudogene in the IRb region. This suggests that the ycf1 pseudogene in the IRb was lost in the evolution of the two species. In the cp genomes of *A. tanguticus* and the Solanaceae species mentioned above, the rps19 gene is located within the LSC/IRb border, except *P. tangutica*, where rps19 exists only in the LSC region. In some species, the 3′ region truncated rps19 pseudogene is in the IRa/LSC border [48]. Similarly, rps19 only appeared in *A. tanguticus* and *S. parviflora* in the IRa region. Our results indicate that the rps19 gene duplication is lost in the IRa/LSC border in the cp genomes of eight Solanaceae species (Figure 4). In the *P. tangutica* cp genome, there was also no rpl22 gene in the LSC region and rpl2 expanded to the IRb/LSC border (Figure 4). Similarly, the rpl2 gene is duplicated in *A. tanguticus* and five other species, while one duplication is lost in *P. tangutica*, *H. niger*, *I. ellipticum*, and *P. peruviana*. In general, our results showed that the IR boundary expansion and contraction differed from the Solanaceae species. 

### 3.5. Comparative Chloroplast Genome Analysis 

The whole cp genome of *A. tanguticus* was compared with five other species (*P. tangutica, H. niger, S. parviflora, N. tabacum*, and *D. stramonium*) using the Mauve program [28]. All cp genomes appeared as locally collinear blocks (Figure 5). The coding proteins and the tRNA and rRNA collinearly blocks are relatively conserved in structures and regions. However, gene loss occurred in the six species, such as 150,000–155,000, in which the *P. tangutica* and *H. niger* cp genomes lost two genes compared with the other four species. In addition, there was no gene rearrangement in Solanaceae genomes (Figure 5); this result is consistent with the other species [48,49]. The CD regions in the cp genomes of these six species were analyzed with MAFFT software and VCFtools was used to calculate the nucleotide variability (π) of each gene (Figure 6, Appendix A). The mean Pi value was 0.006936 among the six species. Similarly, IR regions showed less variability than that of the LSC and SSC regions. Among 113 genes, 26 genes showed no change in the six species, whereas 30 mutation hotspot regions were discovered (Pi > 0.01). These highly divergent proteins included matK, clpP, rps16, rps12, and ycf1, which had the highest Pi value (0.04246) (Figure 6, Appendix A). These regions might be going through rapid nucleotide substitution at the species level, suggesting the possible use of molecular markers for phylogenetic studies and plant identification.

### 3.6. Synonymous (Ks) and Non-Synonymous (Ka) Substitution Rate Analysis 

Synonymous (Ks) and non-synonymous (Ka) were parameters for assessing the rate of gene divergence in a molecular evolution study [30]. To characterize the molecular evolution pattern, we calculated the Ka/Ks ratios of the *A. tanguticus* cp genome and the results were compared with those of five closely related species (*P. tangutica*, *H. niger*, *S. parviflora*, *D. stramonium*, and *N. tabacum*) (Figure 7; Appendix A). Of the 86 protein-coding genes, most had Ka/Ks values < 1. The average ratio was 0.24–0.32 for the five comparison groups. This shows that these species were mostly the outcome of vigorous purifying selection. In the Hyoscyameae tribe, cemA, rpoC2, and clpP exhibited positive selection in *S. parviflora* and *P. tangutica*. In a previous study, the clpP gene showed positive selection in various plant lineages, such as Oenothera [50], and cemA and rpoC2 also appeared to have undergone adaptation evolution in Gossypium and Populus, respectively [51,52]. Our results are consistent with these observations. In a recent study, clpP was shown to degrade misfolded proteins and protected cps from abiotic and biotic stress [53]; loss of this protein may lead to inappropriate structural changes in cps [54]. cemA proteins play an important role in protein sorting signals [55] and rpoC2 is an important subunit of PEP, which is required for transcription of photosynthesis genes and is involved in cp biogenesis [56]. In Nicotiana, positive selection acting on the ATP synthase and NAD(P)H dehydrogenase encoding genes was believed to be an adaptation to novel ecological conditions [57]. Here, our results indicate that protein metabolism may be an important adaptation strategy for *A. tanguticus* and other species in Hyoscyameae. As an endemic species on the Qinghai–Tibetan Plateau, *A. tanguticus* is often exposed to low temperatures, high UV-B radiation, and other abiotic stresses. Degrading misfolded proteins and a sorting signal system may help maintain plasmid function. Therefore, our results suggest that these genes are also important for Hyoscyameae tribe species evolution and adaptation to adverse environments. 

### 3.7. Phylogenetic Analysis of the A. tanguticus cp Genome 

The tribe Hyoscyameae is distributed throughout Eurasia and is considered as a monophyly in Solanaceae [18,19]. However, its genera relationship within this tribe remains unclear. Cp genomes provide whole phylogenetic information, which is suitable for establishing phylogenetic relationships in plants. Whole cp genomes and protein-coding genes of 21 species were utilized to create an ML phylogenetic tree to understand the phylogenetic position of *A. tanguticus* and to further elucidate the evolutionary relationship in the Solanaceae (Appendix A). Most nodes were strongly supported by 100% bootstrap value (Figure 8A). Moreover, *A. tanguticus* was clustered with three species (*P. tangutica*, *H. niger*, and *S. parviflora*) with 100% bootstrap support. The three Hyoscyameae tribe species were clustered together more closely. *D. stramonium*, *N. sylvestris*, and other species were clustered outside this branch with 100% bootstrap support. These findings are consistent with previous studies [18,19,57,58]. In a Hyoscyameae tribe study, *A. tanguticus* grouped with *H. niger* when several chloroplast markers were combined [18,19]. Our results are consistent with this observation. However, when whole genome data are used, *P. tangutica* and *S. parviflora* exhibited a closer relationship in the Hyoscyameae tribe (Figure 8B). This may be due to a data information shortage, such as in *Physochlaina*. Since *A. tanguticus* is an endemic species that grows on the Qinghai–Tibetan Plateau, sequencing, and characterization of additional cp genomes of Hyoscyameae tribe species may provide further insight on the phylogenetic relationships and evolution among these species. 

## 4. Conclusions

High throughput sequencing was used to obtain the whole cp genome of *A. tanguticus*. The cp genome is 152,714 bp and encodes 126 functional genes. Analysis of repeat sequences revealed that forward and palindromic repeats of 40–49 bp were more common in the Hyoscyameae tribe, and 30–39 bp reverse repeats may play a more significant role in *A. tanguticus* and *P. tangutica* cp genome rearrangement and in adapting to extreme conditions on the Qinghai–Tibetan Plateau. The *A. tanguticus* cp genome contained a total of 199 SSRs and can be utilized for the development of molecular markers for phylogenetic and population studies. The genome structure and gene organization were similar to three previously reported species in the Hyoscyameae tribe, while gene contraction and expansion were clearly different in the *A. tanguticus* cp genome. Evolutionary analysis indicated that clpP, cemA, and ropC2 showed positive selection, which suggests that protein metabolism may be an important adaptation strategy for *A. tanguticus* and other species in Hyoscyameae on the Qinghai–Tibetan Plateau. Phylogenetic analysis indicated that *A. tanguticus* has a closer relationship with *P. tangutica*. These results provide significant information for forthcoming studies on phylogenetic and evolutionary adaptation in *A. tanguticus* and its related species. 

## Figures and Tables

**Figure 1 genes-13-02125-f001:**
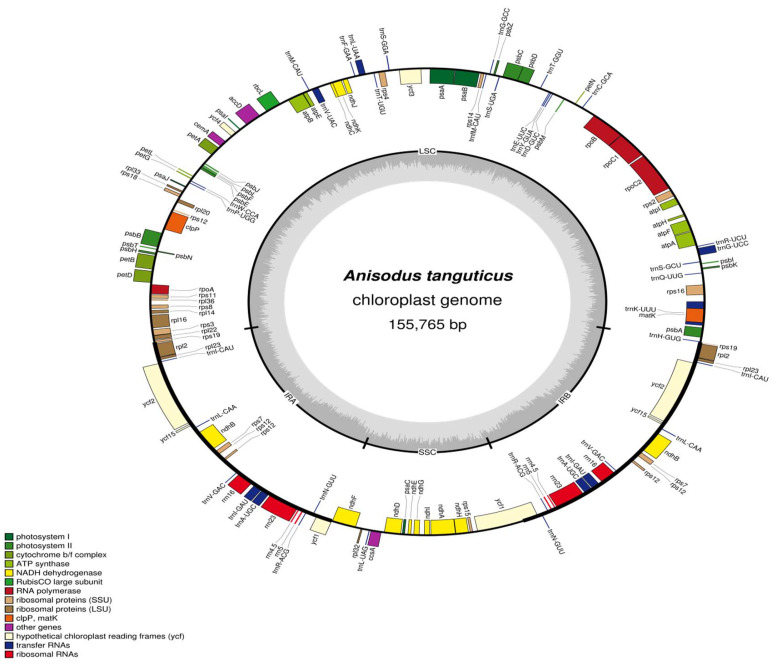
Gene map of the *A. tanguticus* chloroplast genome. (Genes shown outside the outer circle are transcribed clockwise and those insides are transcribed counterclockwise. The colored bars indicated different functional groups).

**Figure 2 genes-13-02125-f002:**
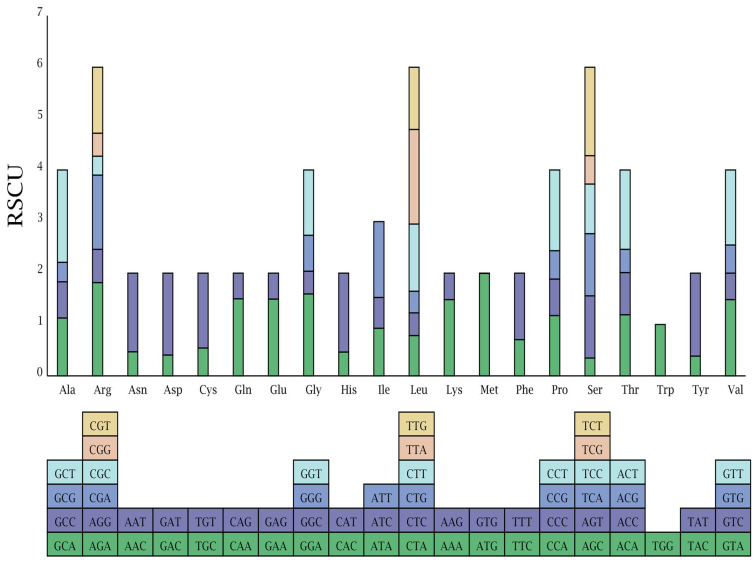
Codon content of 20 amino acids and the stop codon of 81 coding genes of the *A. tanguticus.*

**Figure 3 genes-13-02125-f003:**
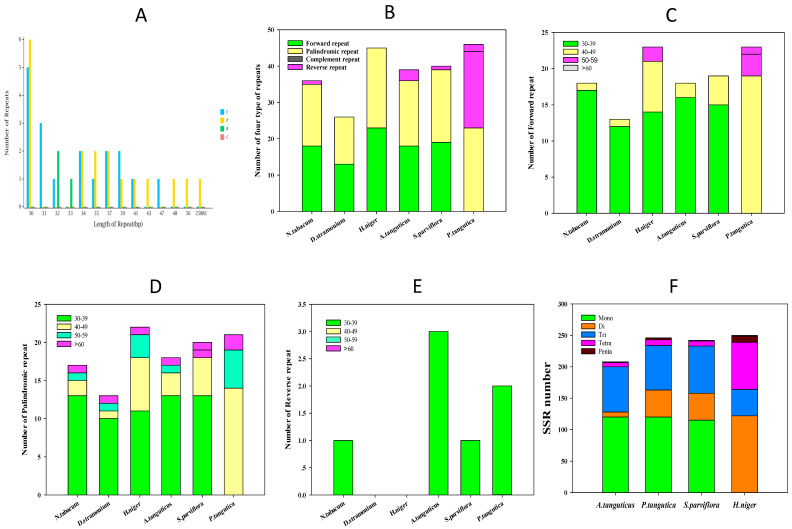
Repeat sequences and SSR analysis of *A. tanguticus* and related species of Solanaceae. (**A**) Profiling of repeat sequences of *A. tanguticus*; (**B**) number of four types of repeat of *A. tanguticus* and five related species; (**C**) comparison of a forward repeat of *A. tanguticus* and five related species; (**D**) comparison of a palindromic repeat of *A. tanguticus* and five related species; (**E**) comparison of a reverse repeat of *A. tanguticus* and five related species; (**F**) SSR number of *A. tanguticus* and three related species.

**Figure 4 genes-13-02125-f004:**
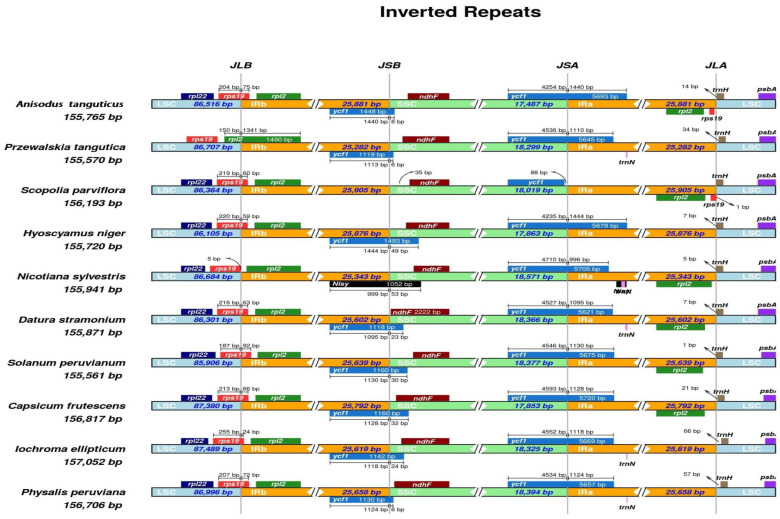
Comparison of the border position of LSC, SSC, and LR regions among *A. tanguticus* and nine species’ chloroplast genomes in Solanaceae. Genes are represented by boxes above (negative strand) and below (positive strand) the proportionate line. The size of each gene and their relative position at the junctions are shown in base pairs (bp).

**Figure 5 genes-13-02125-f005:**
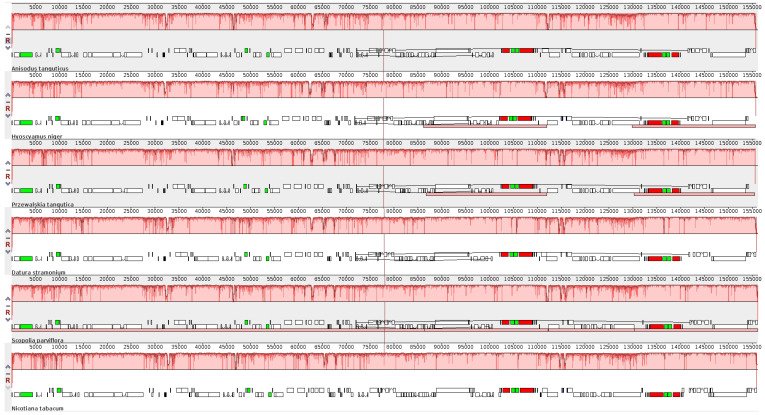
Sequence alignment of four chloroplast genomes in Solanaceae with *A. tanguticus* by Mauve software. Within each of the alignments, local collinear blocks are presented by blocks of the same color connected by lines. Annotations of rRNA, protein-coding, and tRNA genes are shown in red, white, and green boxes, respectively.

**Figure 6 genes-13-02125-f006:**
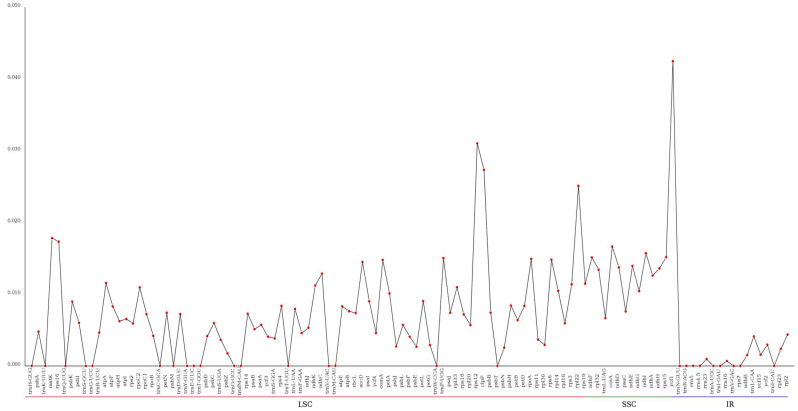
Comparison of nucleotide variability (Pi) among *A. tanguticus* and related species.

**Figure 7 genes-13-02125-f007:**
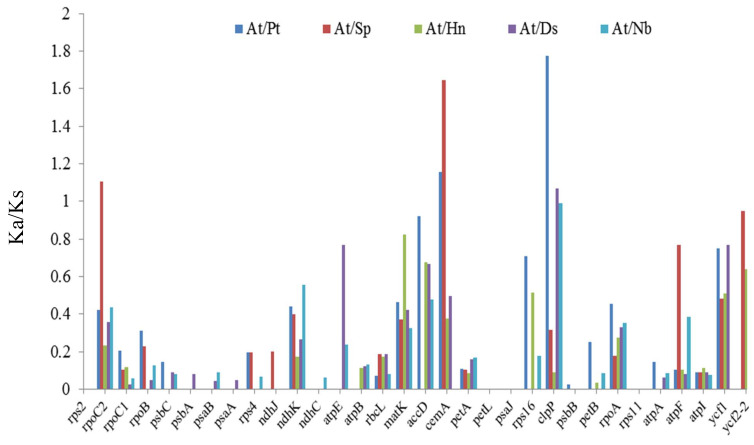
The Ka/Ks ratios of protein-coding genes of *A. tanguticus* and related species.

**Figure 8 genes-13-02125-f008:**
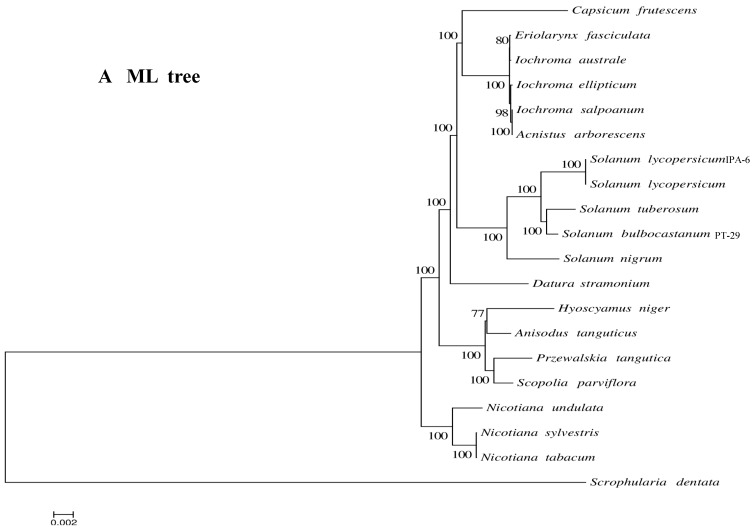
The phylogenetic tree base on 20 cp chloroplast species was constructed with ML methods. Bootstrap values were shown at the nodes. ((**A**). ML tree of coding genes; (**B**). ML tree of whole cp genomes).

**Table 1 genes-13-02125-t001:** Comparison analyses of *A*. *tanguticus* with six close species.

Genome Feature	*A. tanguticus*	*P. tangutica*	*S. parviflora*	*H. niger*	*D. stramonium*	*N. tabacum*	*S. tuberosum*
Genome Size (bp)	155,765	155,569	156,193	155,720	155,871	155, 943	155,298
LSC (bp)	86,516	86,707	86,364	86,105	86,301	86, 686	85,749
SSC (bp)	17,487	18,288	25,905	17,863	18,366	18, 573	18,373
IR (bp)	25,881	25,287	25,876	25,876	25,602	25,342	25,595
GC content (%)	37.63	37.6	37.6	37.6	37.9	37.9	37.9
Total number of genes	132	138	131	118	134	156	130
Protein-coding gene	87	85	86	80	88	111	81
tRNA	37	44	37	30	38	37	37
rRNA	8	8	8	4	8	8	8

**Table 2 genes-13-02125-t002:** List of genes in the cp genome of *A. tanguticus*.

Category for Genes	Groups of Genes	Name of Genes
Photosynthesis	Subunits of photosystem Ⅰ	psaA, psaB, psaC, psaI, psaJ
	Subunits of photosystemⅡ	psbA, psbB, psbC, psbD, psbE, psbF, psbH
		psbI, psbJ, psbK, psbL, psbM, psbN, psbT, psbZ
	Subunits of cytochrome b/f complex	petA, petB, petD *, petG, petL, petN
	Large subunit of Rubisco	rbcL
	Subunits of ATP synthase	atpA, atpB, atpE, atpF *, atpH, atpI
	Subunits of NADH-dehydrogenase	ndhA *, ndhB ^a^*, ndhC, ndhD, ndhE, ndhF, ndhG, ndhHndhI, ndhJ, ndhK
Self-replication	Ribosomal RNA genes	rrn16 ^a^, rrn23 ^a^, rrn5 ^a^, rrn4.5 ^a^
	Transfer RNA genes	*trnA-UGC *^a^*, *trnC-GCA*, *trnD-GUC*, *trnE-UUC *^a^*, *trnF*-*GAA*, *trnfM*-*CAU*, *trnG*-*GCC*, *trnG-UCC **, *trnH*-*GUG*, *trnI-CAU ^a^*, *trnI-GAU ^a^**, *trnK-UUU **, *trnL-CAA ^a^*, *trnL-UAG*, *trnL-UAA *, trnM*-*CAU*, *trnN-GUU ^a^*, *trnP-UGG*, *trnQ-UUG*, *trnR-ACG ^a^*,
		*trnR-UCU*, *trnS-GCU*, *trnS-GGA*, *trnS-UGA*, *trnY-GUA*, *trnT-GGU, trnT-UGU*, *trnV*-*GAC ^a^*, *trnV-UAC **, *trnW*-*CCA*
	Small subunit of ribosome	rps2, rps3, rps4, rps7 ^a^, rps8, rps11, rps12 ^a^**, rps14, rps15, rps16, rps18, rps19
	Large subunit of ribosome	rpl2 ^a^*, rpl14, rpl16 *, rpl20, rpl22, rpl23 ^a^, rpl32, rpl33, rpl36
	DNA-dependent RNA polymerase	rpoA, rpoB, rpoC1 *, rpoC2
Other genes	Maturase	matK
	Envelope membrane protein	cemA
	Subunit of acetyl-CoA	accD
	C-type cytochrome synthesis gene	ccsA
	Protease	clpP **
Genes of unknown function	Conserved Open reading frames	ycf1 ^a^, ycf2 ^a^, ycf3 **, ycf4, ycf15 ^a^

Note: * means the gene contained one intron; ** means the gene contained two introns; ^a^ indicates the number of the repeat unit is 2.

## Data Availability

Data are available from the corresponding author.

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
