# Peer review of "Characterization of the Evolutionary Pressure on Anisodus tanguticus Maxim. with Complete Chloroplast Genome Sequence"

_genes, 2022, doi:10.3390/genes13112125_

Round 1
Reviewer 1 Report
Following are my comments:In this study authors have performed de novo assembly of chloroplast genome and are still managing to assemble a complete genome with all the features is a creditable aspect.
The authors need to further clarify the following sentence from the M&M section “Clean reads were then mapped to the database, which was constructed from all cp genome sequences found in the NCBI based on coverage and similarity”
Figures, especially phylogeny is not visible, authors need to provide high-resolution figures.
Other than this I also suggest authors correct some grammar and italicization errors.
Author Response
The authors need to further clarify the following sentence from the M&M section “Clean reads were then mapped to the database, which was constructed from all cp genome sequences found in the NCBI based on coverage and similarity”
Answer: Yes,plastid reads were filtered by reference mapping to Solanaceae plastid genome sequences using Geneious Prime software.
Figures, especially phylogeny is not visible, authors need to provide high-resolution figures.
Answer: We would provide the high-resolution figures of phylogeny.
Other than this I also suggest authors correct some grammar and italicization errors.
Answer: We correct the grammar and italicization errors.
Reviewer 2 Report
The manuscript "Characterization of the evolutionary pressure on Anisodus tanguticus Maxim. with complete chloroplast genome sequence" is a nice work to be performed.
The title of the manuscript needs some improvement like "Characterization of the evolutionary consequences on Anisodus tanguticus Maxim. based on the complete chloroplast genome sequence".
Abstract Line No. 15. "The cp contains" should be "The cp genome contains".
Abstract Line No. 21. "which differs from" can be written as "in comparison of" or "comparing".
The "Hyoscyameae tribe" have been used at several places in the manuscript as well as in key words, the linkage and importance should be elaborated.
Line No. 44-45. The sentence "considered a vital resource of genetic markers to study the profiling of gene distribution and their molecular phylogenetic relationships" should be revised.
Linke No. 54 The reference should be in one parentheses.
Line No. 56-58 "Despite the development........(Zheng et al., 2008)" Should be re-framed.
There is a lack of consistency of continuity between sentences. Improvement is needed throughout the manuscript.
Line No. 65-66 "There is currently limited cp genome information of A.tanguticus", The authors mean to say partial sequence availability or other full cp genomes are available for A.tanguticus?
Line No. 79-80. Correct it, there is repetition of "DNA".
Explain these Point
1. SOAPdenovo2 has two commands, SOAPdenovo-63mer and SOAPdenovo-127mer. The first
one is suitable for assembly with k-mer values less than 63 bp, requires less memory and
runs faster. The latter one works for k-mer values less than 127 bp...... Authors have not
mentioned the suitable k mer and should also provide the reason for choosing the kmer for
this denovo.
The comment no. 1 pertains to line no 85 to 92 of page no. 2
2. Which pair end they have used for the gap filling.
The comment no. 2 pertains to page no. 2 (line no. 82 and 83)
3. Author should mention the sequencing quality produced during gap filling.
4. The produced sequences should be examined for homozygosity as authors have mentioned it as endangered species.
Comment No. 3 and 4 th comment pertains to line no. 55 to 58 on page no. 2
5. Does author observes some genes that have LoF variations.
The comment no. 5 pertains to conclusion part (line no. 342 and 343)
In the Comparative analysis of cp genomes, have the authors included previously published work with title "The Complete Chloroplast Genome Sequences of Anisodus Acutangulus and a omparison with Other Solanaceae Species" having complete cp genome sequence "Anisodus tanguticus chloroplast, complete genome GenBank: MK347419.1".
https://doi.org/10.1016/j.ccmp.2021.100002
The work is a nice attempt, lacking novelty but should be given an opportunity to revise it in light of the above published work.
Author Response
The manuscript "Characterization of the evolutionary pressure on Anisodus tanguticus Maxim. with complete chloroplast genome sequence" is a nice work to be performed.
The title of the manuscript needs some improvement like "Characterization of the evolutionary consequences on Anisodus tanguticus Maxim. based on the complete chloroplast genome sequence".
Answer: Thank you. We changed the title of the manuscript.
Abstract Line No. 15. "The cp contains" should be "The cp genome contains".
Answer: Thank you. We corrected the error in abstract line No.15.
Abstract Line No. 21. "which differs from" can be written as "in comparison of" or "comparing".
The "Hyoscyameae tribe" have been used at several places in the manuscript as well as in key words, the linkage and importance should be elaborated.
Answer: Yes, we corrected the sentence in abstract line No.21.
Line No. 44-45. The sentence "considered a vital resource of genetic markers to study the profiling of gene distribution and their molecular phylogenetic relationships" should be revised.
Answer: Yes, we revised this sentence as “cp genomes are a vital resource to develop genetic markers to understand the species distribution and their molecular phylogenetic relationships”.
Linke No. 54 The reference should be in one parentheses.
Answer: Yes, we corrected the reference in one parentheses, in line No.54.
Line No. 56-58 "Despite the development........(Zheng et al., 2008)" Should be re-framed.
There is a lack of consistency of continuity between sentences. Improvement is needed throughout the manuscript.
Answer: Thanks for the suggestion, Sir. We rewrite most of the manuscript.
Line No. 65-66 "There is currently limited cp genome information of A.tanguticus", The authors mean to say partial sequence availability or other full cp genomes are available for A.tanguticus?
Answer: Here we mean that there no full cp genomes are available for A.tanguticus in line No.65-66.
Line No. 79-80. Correct it, there is repetition of "DNA".
Answer: Yes, we deleted the second DNA in Line No.79-80.
Explain these Point
- SOAPdenovo2 has two commands, SOAPdenovo-63mer and SOAPdenovo-127mer. The first one is suitable for assembly with k-mer values less than 63 bp, requires less memory and runs faster. The latter one works for k-mer values less than 127 bp......
Authors have not mentioned the suitable k mer and should also provide the reason for choosing the kmer for this denovo.
Answer:Yes, we added this to the method part. We applied SPAdes v3.10.1(http://cab.spbu.ru/software/spades/)to assemble the cp genomes and the two SOAPdenovo2 commands were used in the study, under commander SOAPdenovo-63mer, we performed kmer at 55 and under SOAPdenovo-127mer, kmer was 87 and 121, respectively. Considering the N50 value, we tried these parameters and selected better results for further analysis.
- Which pair end they have used for the gap filling.
Answer: We use the two pair end reads for the gap filling with Gapfiller v2.1.1(https://sourceforge.net/projects/gapfiller/), if some gap couldn’t be filled, we combined the PCR technology to complete.
The comment no. 2 pertains to page no. 2 (line no. 82 and 83)
- Author should mention the sequencing quality produced during gap filling.
Answer: Yes, we completed the sequencing quality produced in the method part. In the study, the Q20 and Q30 value was 97.66% and 93.41% respectively.
- The produced sequences should be examined for homozygosity as authors have mentioned it as endangered species.
Answer: The sample was collected at Haibei station, it was a bit far from Xi’ning. We would check this in the future.
Comment No. 3 and 4 th comment pertains to line no. 55 to 58 on page no. 2
- Does author observes some genes that have LoF variations.
The comment no. 5 pertains to conclusion part (line no. 342 and 343)
Answer: Yes, we observed the gene such rps19 in the cp genome was LoF.
In the Comparative analysis of cp genomes, have the authors included previously published work with title "The Complete Chloroplast Genome Sequences of Anisodus Acutangulus and a omparison with Other Solanaceae Species" having complete cp genome sequence "Anisodus tanguticus chloroplast, complete genome GenBank: MK347419.1".
https://doi.org/10.1016/j.ccmp.2021.100002
Answer: Thank you. in this study, we target our genome with other Solanaceae members rather than target within the same genus. We have sequenced more genomes in the Anisodus sequence and in the future, we will publish that paper within genus studies in detail.